# *SIRT7* Acts as a Guardian of Cellular Integrity by Controlling Nucleolar and Extra-Nucleolar Functions

**DOI:** 10.3390/genes12091361

**Published:** 2021-08-30

**Authors:** Poonam Kumari, Shahriar Tarighi, Thomas Braun, Alessandro Ianni

**Affiliations:** Department of Cardiac Development and Remodeling, Max-Planck-Institute for Heart and Lung Research, Ludwigstrasse 43, 61231 Bad Nauheim, Germany; Poonam.Kumari@mpi-bn.mpg.de (P.K.); Shahriar.Tarighi@mpi-bn.mpg.de (S.T.); Thomas.Braun@mpi-bn.mpg.de (T.B.)

**Keywords:** sirtuins, *SIRT7*, deacetylation, stress responses, nucleolus

## Abstract

Sirtuins are key players for maintaining cellular homeostasis and are often deregulated in different human diseases. *SIRT7* is the only member of mammalian sirtuins that principally resides in the nucleolus, a nuclear compartment involved in ribosomal biogenesis, senescence, and cellular stress responses. The ablation of *SIRT7* induces global genomic instability, premature ageing, metabolic dysfunctions, and reduced stress tolerance, highlighting its critical role in counteracting ageing-associated processes. In this review, we describe the molecular mechanisms employed by *SIRT7* to ensure cellular and organismal integrity with particular emphasis on *SIRT7*-dependent regulation of nucleolar functions.

## 1. Mammalian Sirtuins: General Functions and Activation in Response to Stress

Sirtuins are highly conserved enzymes that principally act as NAD^+^-dependent histone/protein deacetylases although some members also possess mono-ADP ribosylation and other less characterized activities [1,2,3,4,5,6]. For deacetylation, sirtuins catalyze the transfer of the acetyl group from the substrate to NAD^+^, concomitant with the release of 2’–O-acetyl-ADP-ribose and nicotinamide (NAM). During mono-ADP ribosylation, the ADP-ribose (ADPR) is transferred from NAD^+^ to the target protein and NAM is released (Figure 1a). Interestingly, NAM is a potent inhibitor of sirtuins, suggesting the presence of a finely controlled negative auto-regulatory loop, modulating the enzymatic activity (Figure 1a). In mammals, seven sirtuin members (*SIRT1*–*SIRT7*) have been described. These molecules share a conserved catalytic domain but differ substantially in their N-terminal and C-terminal sequences, which are fundamental for interactions with specific targets as well as for the specification of subcellular localization. Sirtuins are found in different cellular compartments: *SIRT1*, *SIRT6*, and *SIRT7* are principally localized in the nucleus while other sirtuins reside in the mitochondria or in the cytoplasm as illustrated in Figure 1b. *SIRT7* is the only member of the family that is highly enriched in the nucleolus due to the presence of nuclear and nucleolar localization sequences at the N-terminal and C-terminal regions [7] (Figure 1b). Sirtuins efficiently shuttle between different compartments in response to numerous intracellular and extracellular stimuli to promote the activation of specific signaling pathways. These molecules activate a complex network of cellular responses to stress to maintain cellular homeostasis, including regulation of the transcription of specific target genes, modulation of chromatin structure, activation of mechanisms of DNA repair, metabolic adaptation to stress, and modulation of apoptosis among others [8] (Figure 1c). The ability of sirtuins to control such a broad range of biological functions mainly derives from their capacity to control both chromatin-related and -unrelated targets. Sirtuins act as potent epigenetic silencers by promoting heterochromatin formation through the direct deacetylation of different histone marks or indirectly by controlling the activity of numerous histone modifiers such as methytransferases and acetyltransferases [8,9]. Notwithstanding, sirtuins control the activity and functions of enzymes, transcription factors, and other chromatin unrelated molecules mainly through direct deacetylation, highlighting their complex functions in the regulation of cellular processes [3].

Due to their critical role in modulating stress related cellular reactions, sirtuins act as key anti-ageing molecules in low eukaryotes as well as in mammals and are assumed to promote life span extension [6]. Consistently, age-dependent decline in sirtuins expression and/or activity has been associated with the onset of severe aging-associated diseases such as cardiovascular diseases [3,10], diabetes [11], inflammation [12], neurological disorders [13], and cancer [14].

Different mechanisms are employed to promptly activate sirtuins following stress. Stressors modulate sirtuin genes’ expression, protein stability, or control their binding to specific inhibitors [5] (Figure 1c). Moreover, the strict dependency of the catalytic activity of sirtuins on NAD^+^ allows swift adaption to the metabolic state of cells. Stress conditions such as glucose deprivation or caloric restriction (CR), which increase NAD^+^ levels due to enhanced mitochondrial respiration and dramatically induce sirtuins expression and/or activity [6] (Figure 1c). CR represents the most prominent intervention to delay ageing and extend life span in different experimental organisms [15]. Different studies demonstrated that the beneficial effects of CR can be attributed to the activation of sirtuins, at least in part [16]. Thus, the activation of sirtuins might improve cellular adaptation to stress, induced by nutrient deprivation.

The acquisition of specific post-translational modifications (PTMs) such as phosphorylation, methylation, and ubiquitination, among others, represents another mechanism to control the activity of sirtuins. PTMs control the interaction of sirtuins with specific targets, their subcellular localization and catalytic activity, in some cases by interfering with the capacity to bind NAD^+^ [17,18,19]. Intriguingly, recent studies also demonstrated that mammalian sirtuins possess auto-regulatory mechanisms. For instance, *SIRT1* is capable of auto-catalytic activation by auto-deacetylation [20] while *SIRT7* possesses auto-mono-ADP ribosylation activity [1]. Interestingly, cross-regulation between mammalian sirtuins is also employed to fine-tune their functions. The binding of *SIRT7* to *SIRT1* inhibits *SIRT1* auto-catalytic activation resulting in the stimulation of adipogenesis and destabilization of constitutive heterochromatin [9,20]. In sharp contrast, binding of *SIRT1* to *SIRT7* is fundamental to promote the formation of ribosomal DNA heterochromatin and to repress E-cadherin expression, thus stimulating cancer metastasis [21,22]. Additionally, a synergistic effect between *SIRT1* and *SIRT6* has been described. *SIRT1*-mediated deacetylation of *SIRT6* is fundamental for recruitment of *SIRT6* to double strand breaks (DSBs), thereby facilitating chromatin remodeling required for efficient DNA repair [23]. These findings illustrate the complexity of the regulation of sirtuins under physiological and pathological conditions. More research is warranted to unveil the molecular mechanisms controlling the regulation of these molecules in stress induced cellular processes, in order to identify novel pathways suitable for therapeutic interventions in diseases characterized by alterations in functions of sirtuins (Figure 1c).

## 2. *SIRT7* Controls Multiple Functions of the Nucleolus

### 2.1. SIRT7 Stimulates rDNA Transcription and Ribosomes Biogenesis

The nucleolus is the cellular compartment responsible for ribosomal DNA (rDNA) transcription and ribosomes biogenesis. This membrane-less organelle is arranged around actively transcribed rDNA genes as a result of the recruitment of RNA binding proteins, ribosomal proteins, and other molecules involved in ribosome biogenesis with the nascent ribosomal RNA (rRNA) [24]. The assembly of the ribosomes is a highly regulated process that is tightly controlled in response to nutrient availability and other stimuli. The ribosomal rRNA is initially transcribed as a precursor pre-rRNA, which is then cleaved into the mature 28S, 18S, and 5.8S rRNAs. Small nucleolar ribonuclear proteins (snoRNPs) and small nucleolar RNAs together with other enzymes are required for efficient cleavage of the pre-rRNA and/or to catalyze the acquisition of critical post-transcriptional modifications. These events are indispensable for proper folding of rRNAs and their assembly with ribosomal proteins to generate the mature ribosome subunits [25,26].

The enriched localization of *SIRT7* in the nucleoli prompted different groups to investigate its biological functions in this nuclear compartment. *SIRT7* emerged to be a critical factor that promotes rRNA transcription and maturation by acting at different levels. *SIRT7* stimulates rDNA transcription by facilitating the recruitment of RNA polymerase I (Pol I) at rDNA genes both through its interaction with the transcription factor UBF1 and deacetylation of the Pol I subunit PAF53 [27,28]. Additionally, *SIRT7* deacetylates the nucleolar protein fibrillarin (FBL), facilitating FBL-mediated histone methylation that is essential to activate rDNA transcription during interphase and its resumption at the end of mitosis (Figure 2 and Table 1) [29,30]. Besides stimulating rDNA transcription, *SIRT7* facilitates pre-rRNA processing through deacetylation of U3–55k, a core component of the U3 snoRNP complex [31] (Figure 2 and Table 1). In sharp contrast, recent work demonstrated that *SIRT7* epigenetically represses expression of specific ribosomal proteins, indicating a highly complex role of this molecule in the maintenance of ribosome homeostasis [32,33]. *SIRT7* is also a critical regulator of RNA Pol III mediated transcription and stimulates expression of different tRNAs, suggesting that the capacity of *SIRT7* to activate global protein biosynthesis employs additional mechanisms besides promoting ribosome biosynthesis [34].

Sustained ribosomes and protein biosynthesis are required for efficient cellular growth and are prominently enhanced in metabolically active tissues. Consistently, *SIRT7* levels are elevated in highly proliferative tissues but are dramatically reduced in post-mitotic cells [27]. *SIRT7* acts as a potent oncogene in different malignancies and is upregulated in numerous human cancers [35]. The *SIRT7*-mediated stimulation of ribosomes biogenesis is considered fundamental for *SIRT7*-dependent oncogenic functions, although other mechanisms are involved too [32,33,36]. In addition, the ability of *SIRT7* to epigenetically suppress the expression of specific ribosomal proteins and the consequent alteration of the translation machinery has been proposed as an additional mechanism utilized by *SIRT7* to contribute to oncogenic transformation [32].

### 2.2. SIRT7 Ensures Genomic Stability by Securing rDNA Repeats Integrity: A Link to Ageing?

Besides its well-characterized role in ribosomes biogenesis, the nucleolus has been recognized as a critical compartment involved in maintaining genomic stability, preventing ageing, and enabling cellular stress responses [37].

To fulfil metabolic requirements, cells possess multiple copies of head-to-tail arranged rDNA repeats that constitute the nucleolar organizer regions (NORs) [38]. However, at a given time, only a subset of rDNA genes is actively transcribed while roughly half of the rDNA repeats in mice are kept in a highly silenced state through formation of compact heterochromatin [39]. Due to their repetitive nature, the maintenance of proper compact heterochromatin at the rDNA repeats is fundamental to prevent homologous recombination. In yeasts, recombination of rDNA repeats and their exclusion from the genome results in the formation of extrachromosomal rDNA circles (ERCs) [40]. The redistribution of excised ERCs repeats throughout the nucleolus promotes the formation of ectopic nucleoli, which gives rise to a typical phenotype often referred to as “fragmented nucleolus”. The accumulation of ERCs and appearance of a fragmented nucleolus is considered to promote premature ageing in yeasts [40]. However, the exact mechanisms by which rDNA instability promotes ageing remains largely unknown. It was proposed that the accumulation of ERCs may bind and neutralized factors involved in the maintenance of a “young” cellular phenotype [41], although further studies demonstrated that rDNA instability per se can cause premature ageing, independently of ERCs accumulation, by promoting activation of the DNA damage response leading to cellular senescence [42,43]. The importance of maintenance of rDNA repeats and nucleolar integrity as a mechanism to prevent cellular senescence has also been proposed in mammals. Loss of rDNA genes occurs in humans during ageing [44,45] and several human diseases associated with genomic instability and signs of premature ageing show alterations or instability of rDNA repeats [37,40,46]. Sirtuins play a critical role in maintaining the stability of rDNA repeats. In yeasts, the sirtuin homologue SIR2 is critical to stabilize the number of rDNA repeats and to prevent the accumulation of ERCs. Consequently, ablation of SIR2 shortens life span while the expression of extra copies of SIR2 delays ageing [40].

Recent studies demonstrated that the function of sirtuins in the maintenance of rDNA stability is conserved in mammals [22,47]. The ablation of *SIRT7* expression results in reduction of rDNA heterochromatin, loss of rDNA repeats, fragmentation of the nucleolus both in vitro and in vivo, and enhanced cellular senescence [22,47]. Intriguingly, *SIRT7* deficient mice show premature signs of ageing and display enhanced genomic instability, suggesting that *SIRT7* counteracts ageing by stabilizing rDNA chromatin [22,48,49]. Mechanistically, *SIRT7* facilitates the formation of a condensed heterochromatin structure at rDNA repeats, which prevents homologous recombination, through different mechanisms. *SIRT7* recruits *SIRT1* at the rDNA genes and promotes *SIRT1*-dependent histone deacetylation [22]. In addition, *SIRT7* enhances heterochromatin formation by facilitating the recruitment of DNA methyltransferase (DNMT1) and the nucleolar remodeling complex (NoRC) to rDNA repeats [22,47] (Figure 2 and Table 1). Furthermore, *SIRT7* forms a molecular complex with SUV39H1, a methyltransferase responsible for the deposition of H3K9 methylation and heterochromatin formation [9]. Interestingly, SUV39H1 together with *SIRT1*, is a component of the energy-dependent nucleolar silencing complex (e-NOSC). E-NOSC contributes to heterochromatin formation and rDNA silencing during nutrient deprivation through *SIRT1*-dependent histone deacetylation and SUV39H1-mediated histone methylation [50]. The depletion of the SUV39H1 homologue in *Drosophila melanogaster* results in heterochromatin relaxation, rDNA instability and formation of ERCs, which is associated with the appearance of fragmented nucleoli [51]. Since the association of SUV39H1 to rDNA genes requires recruitment of *SIRT1* and ablation of *SIRT7* impairs *SIRT1* binding at these chromosomal loci [22], it is reasonable to assume that the inactivation of *SIRT7* will result in global reduction of e-NOSC recruitment at rDNA genes and subsequent loss of heterochromatin at these loci. However, further studies are required to support this claim. Thus, *SIRT7* appears to exert different functions at active and inactive rDNA repeats. At inactive genes, *SIRT7* promotes heterochromatin formation, thus maintaining their stability. At active rDNA repeats, *SIRT7* stimulates Pol I-mediated rDNA transcription [47]. The simultaneous control of these opposite mechanisms by *SIRT7* is surprising, since maintenance of rDNA repeats stability and rDNA transcription appear to be two distinct and unrelated events [52].

In a recent study, it was demonstrated that *SIRT7* is involved in resolving R-loops inside and outside the nucleolus to safeguard genomic stability [53]. R-loops are stable hybrids between nascent RNA and DNA that form during replication. If not resolved, R-loops may cause stalling of the RNA polymerase and ultimately lead to formation of double strand breaks (DSBs) [38,53]. The high transcription rate of rDNA genes predisposes these loci to a dramatic accumulation of R-loops. Interestingly, the accumulation of R-loops has been connected with disrupted nucleolar architecture and appearance of a fragmented nucleolus in mammalian cells [54]. Hence, *SIRT7* may control rDNA and global genomic stability by reducing R-loops accumulation within the nucleolus (Figure 2 and Table 1).

Taken together, *SIRT7* is a critical factor for securing rDNA stability, which may be a crucial mechanism to prevent cellular senescence and extend life span. The pharmacological stimulation of *SIRT7* appears promising for development of future anti-ageing therapies, although further research is required to substantiate this claim.

### 2.3. SIRT7 Triggers the Nucleolar Stress Response to Promote p53 Stabilization in Response to Specific Stress Stimuli

The nucleolus has been recognized as a critical sensor of numerous stressors and is a key player for activation of cellular stress response. Numerous molecules are stored in the nucleolus under physiological conditions. In response to stress, the nucleolus undergoes a profound reorganization leading to a release of these proteins into other cellular compartments, thus allowing their interaction with specific downstream targets and the activation of distinct signaling pathways. This phenomenon is collectively known as the nucleolar stress response (NSR) [55,56]. Compared to the initiation of gene expression, rapid mobilization of nucleolar factors following stress represents a means to promptly activate cellular responses such as cell cycle arrest, DNA repair, modulation of metabolic pathways and apoptosis to ensure cellular homeostasis [57].

The best-characterized consequence of NSR is the rapid stabilization of the tumor suppressor p53. The induction of p53 levels following genotoxic stress is fundamental to ensure cell cycle inhibition, DNA repair or to induce apoptotic death of highly damaged cells. These mechanisms limit the propagation of cells that have acquired mutations and are thereby prone to oncogenic transformation. Without cellular stress, the tumor suppressor p53 is maintained at low levels, which is mainly achieved by the ubiquitin ligase MDM2 that promotes ubiquitination and the subsequent proteasomal degradation of p53. Induction of the NSR induces release of nucleolar proteins (most prominently nucleophosmin; NPM) as well as ribosomal proteins from maturing ribosomes. These molecules associate to MDM2 and disrupt binding of MDM2 to p53, thus promoting p53 stabilization [58]. Recent studies demonstrated that *SIRT7* plays a fundamental role in activating this mechanism under specific stress conditions. In response to ultraviolet (UV)-induced genotoxic stress, *SIRT7* enzymatic activity increases as a consequence of phosphorylation by the kinase ATR, a major player in the DNA damage response. Activated *SIRT7* efficiently deacetylates its nucleolar target NPM favoring translocation of NPM from nucleoli to the nucleoplasm. Deacetylated, nucleoplasmic NPM binds to MDM2, prevents MDM2-dependent p53 degradation, and thus causes the rapid accumulation of p53 and enhanced p53-dependent cell cycle arrest [17] (Figure 3 and Table 1).

The translocation of *SIRT7* in response to glucose starvation represents another mechanism to stabilize p53. Nucleoplasmic *SIRT7* associates and deacetylates the acetyl transferase p300/CBP-associated factor (PCAF), which increases the binding of PCAF to MDM2 favoring MDM2 degradation, p53 stabilization, and cell cycle inhibition [59] (Figure 3 and Table 1). Apparently, distinct mechanisms are employed by *SIRT7* to inhibit MDM2 for promoting p53 stabilization, depending on the type of stress [17,59]. On the other hand, *SIRT7* is not always required for p53 stabilization. For example, *SIRT7* has no impact on p53 stabilization following inhibition of rDNA transcription [17] and rather promotes p53 stabilization in response to specific stress stimuli although through still poorly characterized mechanisms [60]. The capacity of *SIRT7* to directly deacetylate p53 and suppress its transcriptional activity adds another layer of complexity into the mechanisms by which *SIRT7* controls the p53 pathway. Although some studies demonstrated that the deacetylation of p53 by *SIRT7* suppresses its pro-apoptotic functions and favors cell survival under stress [61,62], other studies failed to identify p53 as a deacetylation target of *SIRT7* under particular stress conditions or in vitro [32,59,63]. These apparently contradictory results suggest a highly complex role of *SIRT7* in controlling the p53 pathway that warrants further research.

The induction of the NSR has been proposed as a novel target for anti-cancer therapies due to stimulation of p53 activity [64]. Further characterization of the mechanisms employed by *SIRT7* to control p53 may be crucial for the development of novel pharmacological approaches to treat cancer.

### 2.4. Stress Signals Promote Exclusion of SIRT7 from Nucleoli: A Way to Inhibit rDNA Transcription

The exclusion of *SIRT7* from nucleoli in response to different stress signals has been amply documented, indicating that exclusion of *SIRT7* per se represents a hallmark of the NSR [36,65]. Several lines of evidence suggest a dynamic role of *SIRT7* in the stressed nucleolus that might be employed to fine-tune activation of distinct cellular responses. In the early phase of the NSR, *SIRT7* activates specific signaling pathways through deacetylation of nucleolar targets such as NPM [17]. The exclusion of *SIRT7* from the nucleolus seems to represent a mechanism that prevents hyper-activation of nucleolar *SIRT7* targets. Notwithstanding, the exit of *SIRT7* from the nucleolus results in the hyperacetylation of other nucleolar proteins and consequent modulation of their functions, which may occur during a late phase of the NSR [36]. Furthermore, relocation of *SIRT7* into other cellular compartments is instrumental for rapid activation of extra-nucleolar functions of *SIRT7*.

Biosynthesis of ribosomes is the most highly energy-demanding process that takes place in the cell. Thus, adaptation of this process to nutrient availability is fundamental to prevent unnecessary energy expenditure and to ensure maintenance of cellular homeostasis and survival. [50]. The inability to adapt ribosome biogenesis to external cues has been proposed as a critical mechanism that facilitates ageing in different organisms [46]. In lower eukaryotes, sirtuins play a critical role in rDNA silencing following nutrient starvation [46], while the nuclear sirtuins *SIRT1* and *SIRT7* play opposite roles in this process in mammals. As mentioned above, *SIRT1* is a component of the e-NOSC complex: e-NOSC senses elevated NAD+ levels induced by nutrient deprivation through the activation of *SIRT1*, inducing epigenetic silencing of rDNA genes in a *SIRT1*- and SUV39H1-dependent manner [50]. In sharp contrast, *SIRT7* stimulates rDNA transcription and its exclusion from nucleoli following nutrient starvation is fundamental to ensure the rapid inhibition of rDNA transcription [36]. The exclusion of *SIRT7* from the nucleoli favors hyperacetylation of the RNA Polymerase I subunit PAF53 with subsequent reduction of RNA Pol I recruitment at rRNA genes [28]. In addition, the absence of *SIRT7* in the nucleolus reduces pre-rRNA processing due to hyperacetylation of U3–55k [31]. Taken together, *SIRT7* employs several mechanisms to reduce ribosomes biogenesis in response to nutrient stress [36] (Figure 4 and Table 1).

The mechanisms controlling *SIRT7* translocation from the nucleoli following stress stimuli remain poorly characterized. Maintenance of *SIRT7* in the nucleoli is strictly correlated to rRNA transcription, since its inhibition excludes *SIRT7* from this compartment [27]. Several different pathways inhibit rDNA transcription in response to stress. Thus, the reduction of rRNA expression itself may cause exclusion of *SIRT7* from nucleoli, which will further reinforce transcriptional inhibition, robustly preventing ribosome biogenesis. Post-translational modifications also control the subcellular localization of *SIRT7*. The phosphorylation of *SIRT7* by AMPK (5’ adenosine monophosphate-activated protein kinase), a key molecule for restoring cellular energy levels, favors exclusion of *SIRT7* from nucleoli, resulting in dramatic downregulation of rDNA transcription and energy saving following glucose starvation [66]. *SIRT7* is also phosphorylated by the kinase ATR following genotoxic stress, although the effect of this post-translational modification on the subcellular distribution of *SIRT7* and its contribution to rRNA transcription was not determined [17]. The complex network employed to efficiently exclude *SIRT7* from the nucleolus has been only partially characterized, despite its paramount importance for inhibiting rDNA transcription and regulating other cellular functions (Figure 4 and Table 1).

## 3. *SIRT7* Controls Extra-Nucleolar Functions to Ensure Cellular Integrity Following Stress

In addition to the control of critical nucleolar functions, *SIRT7* is involved in numerous cellular reactions that take place outside this compartment. We assume that the nucleolus acts as a reservoir of *SIRT7* and ensures its rapid mobilization following stress to achieve a robust activation of *SIRT7*-mediated extra-nucleolar functions. For instance, *SIRT7* controls the activation of a transcriptional program that facilitates adaptation to starvation. This process requires auto-mono-ADP ribosylation of *SIRT7*. Auto-modified *SIRT7* binds to the histone variant mH2A1.1, which binds mono-ADP-ribose. Thereby, *SIRT7* is recruited to intragenic regions where it controls expression of nearby genes by modulating chromatin organization [1]. *SIRT7* is also involved in maintaining mitochondria homeostasis. Mitochondria are organelles responsible for cellular energy generation through the biosynthesis of ATP but also control other cellular activities. Dysregulation of mitochondrial homeostasis has been implicated in the development of numerous human diseases including ageing. Moreover, the adaptation of mitochondrial functions in response to nutrient deprivation is imperative for the maintenance of cellular homeostasis and to ensure cell survival [67]. Ablation of *SIRT7* in mice correlates with the onset of cellular and organismal alterations associated with mitochondrial dysfunctions such as accelerated ageing, deterioration of hematopoietic stem cells, cardiac and hepatic diseases, and age-associated hearing loss [68,69]. *SIRT7* employs different mechanisms to control mitochondrial functions. In particular, *SIRT7* acts as a potent epigenetic suppressor of nuclear encoded genes responsible for mitochondria biogenesis [69,70] and represses thereby the mitochondrial protein folding stress response (PFS^mt^). Inhibition of *SIRT7* enzymatic activity under physiological conditions through PRMT6-mediated methylation ensures efficient stimulation of mitochondria biogenesis [70]. Following glucose starvation, binding of *SIRT7* to PRMT6 is disrupted via a mechanism requiring AMPK-dependent phosphorylation, which stimulates *SIRT7*-mediated epigenetic repression of target genes leading to inhibition of de novo biosynthesis of mitochondria [70]. In addition, *SIRT7*-dependent inhibition of the PFS^mt^ ensures cell survival following nutrient deprivation [69]. On the other hand, *SIRT7* stimulates expression of genes controlling mitochondrial functions (such as components of the respiratory chain) by activating the transcription factor GABPβ1 through direct deacetylation [68]. Interestingly, this mechanism has been proposed to play a critical role in mitochondrial homeostasis and metabolic adaptation to nutrient starvation [68]. How the opposing functions of *SIRT7* for expression of mitochondrial genes orchestrate the global adaptation of mitochondria to nutrient deprivation remains to be further characterized. A potential explanation for the opposing effects of *SIRT7* on mitochondria might be that *SIRT7* reduces de novo biosynthesis of mitochondria to prevent energy expenditure in response to stress but at the same time stimulates higher ATP production from the already existing mitochondria, thereby guaranteeing an efficient energy supply (Figure 4). Taken together the available data indicate that translocation of *SIRT7* from nucleoli contributes to cellular and organismal homeostasis in response to stress induced by nutrient deprivation not only by inhibiting rDNA transcription but also by controlling other pathways and organelles, such as mitochondria.

Different studies demonstrated that *SIRT7* acts as a key player in the maintenance of genomic stability following genotoxic stress [4,49,71]. *SIRT7* is recruited to sites of DNA damage where it facilitates DNA repair by modulating the chromatin structure through deacetylation of H3K18 and desuccinylation of H3K122 [4,49]. Moreover, *SIRT7* binds more efficiently in response to genotoxic stress to the kinase ATM (Ataxia telangiectasia mutated), a key molecule involved in the DNA damage response. Binding of *SIRT7* to ATM promotes ATM deacetylation and deactivation, which is a requisite for efficient DNA repair [72]. The exclusion of *SIRT7* from the nucleoli as well as modulation of its catalytic activity following genotoxic stress may be an efficient means to promote rapid activation of DNA repair mechanisms and to attenuate oncogenesis [17,65] (Figure 4). Finally, it is worth to mention that *SIRT7* has been implicated in the activation of cellular responses to hypoxia and endoplasmic reticulum stress [73,74]. Thus, the translocation of *SIRT7* from the nucleolus orchestrates a broad range of adaptive mechanisms to stress to maintain cellular integrity (Figure 4).

## 4. Conclusions

*SIRT7* remains the least characterized member of mammalian sirtuins. However, new experimental evidence supports a critical role of this molecule in the maintenance of genomic stability and global organismal homeostasis. *SIRT7* acts as a key anti-ageing molecule by controlling critical functions of the nucleolus such as the stability of rDNA repeats, activation of the nucleolar stress response, and modulation of rDNA transcription. The nucleolus also represents a prominent storage site for *SIRT7*, ensuring its rapid availability in response to different stressors. Further identification of molecular targets of *SIRT7* as well as the mechanisms governing its translocation from the nucleoli might provide a starting point for the developing a new class of anti-ageing and anti-cancer drugs.

## Figures and Tables

**Figure 1 genes-12-01361-f001:**
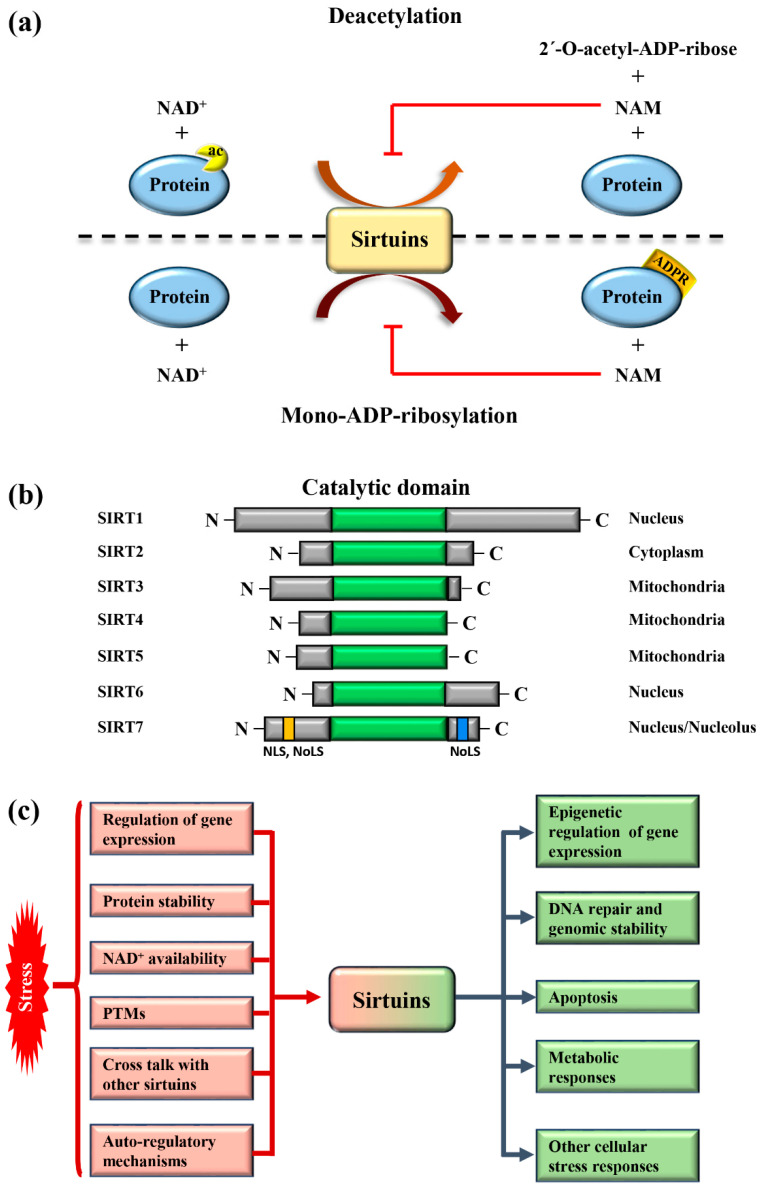
Mammalian sirtuins: general functions and activation in response to stress (**a**) Schematic representation of the main enzymatic activities of mammalian sirtuins. Sirtuins catalyze deacetylation of protein targets by transferring the acetyl group to NAD+, concomitant with the release of nicotinamide (NAM) and 2′-O-acetyl-ADP-ribose (upper panel). In the mono-ADP-ribosylation reaction, the ADP-ribose (ADPR) is transferred from NAD^+^ to the substrate, leading to release of NAM (lower panel). NAM is a potent inhibitor of the enzymatic activity of sirtuins. (**b**) Schematic representation of the structure and subcellular distribution of mammalian sirtuins. *SIRT7* nuclear and nucleolar localization sequences (NLS and NoLS, respectively) are indicated. (**c**) Scheme depicting the mechanisms involved in sirtuins activation following stress (red) and their main biological functions (green).

**Figure 2 genes-12-01361-f002:**
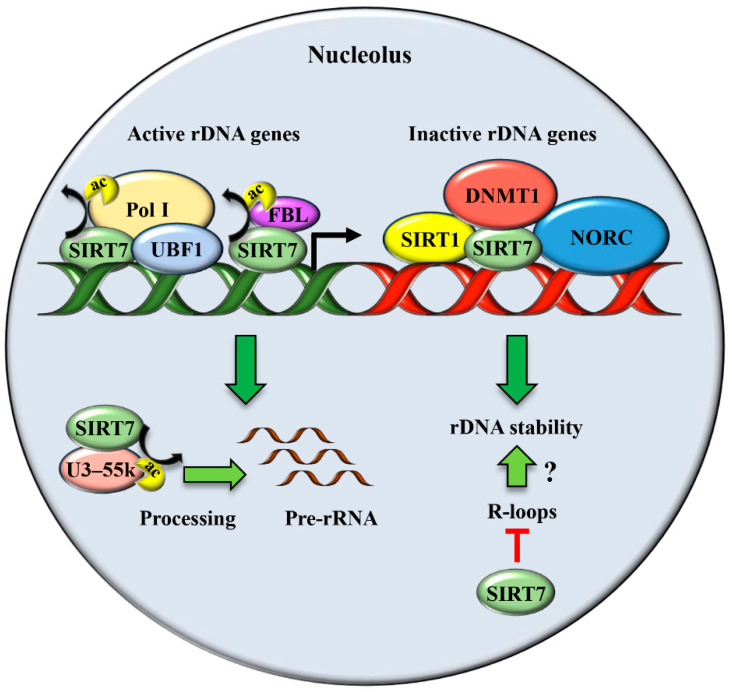
*SIRT7* has a dual function in the unstressed nucleolus. *SIRT7* stimulates rDNA transcription by facilitating recruitment of Pol I both by interacting with UBF1 and through direct deacetylation of the Pol I subunit PAF53. In addition, *SIRT7* deacetylates fibrillarin (FBL) and thereby favors FBL-mediated chromatin remodeling required for stimulation of rDNA transcription. *SIRT7* also promotes pre-rRNA processing by deacetylating U3–55k, a core component of the U3 snoRNP complex. *SIRT7* is a key player for maintaining rDNA stability at inactive rDNA genes by promoting heterochromatin formation through recruitment of *SIRT1*, DNMT1, and the chromatin remodeling complex (NORC). Moreover, *SIRT7* might maintain rDNA stability by facilitating resolution of R-loops.

**Figure 3 genes-12-01361-f003:**
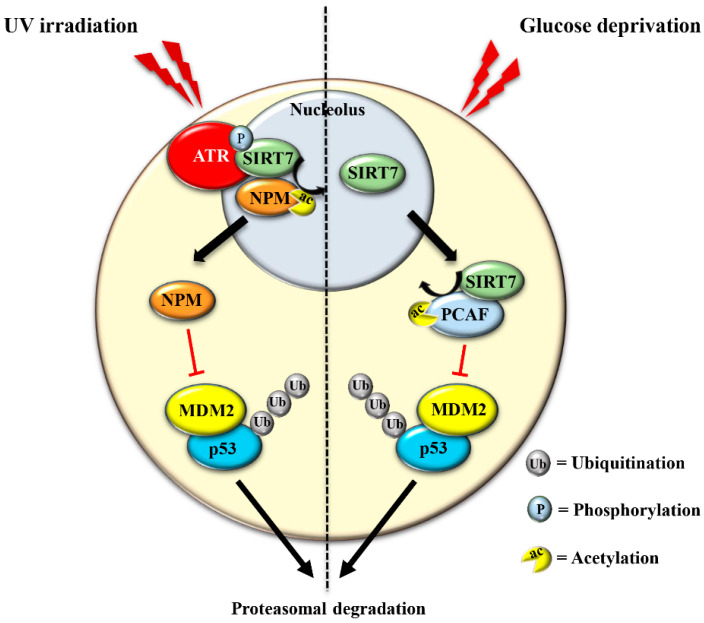
*SIRT7* promotes p53 stabilization in response to distinct stressors. In response to UV-induced stress, *SIRT7* is activated by ATR-mediated phosphorylation. Activated *SIRT7* efficiently deacetylates its nucleolar target nucleophosmin (NPM), facilitating its exclusion from nucleoli. Deacetylated NPM binds and inhibits the ubiquitin ligase MDM2, thereby preventing MDM2- dependent ubiquitination and subsequent proteasomal degradation of p53 (left panel). In response to glucose starvation, *SIRT7* translocates from the nucleolus, associates and deacetylates PCAF, which favors PCAF-mediated degradation of MDM2, leading to p53 stabilization (right panel).

**Figure 4 genes-12-01361-f004:**
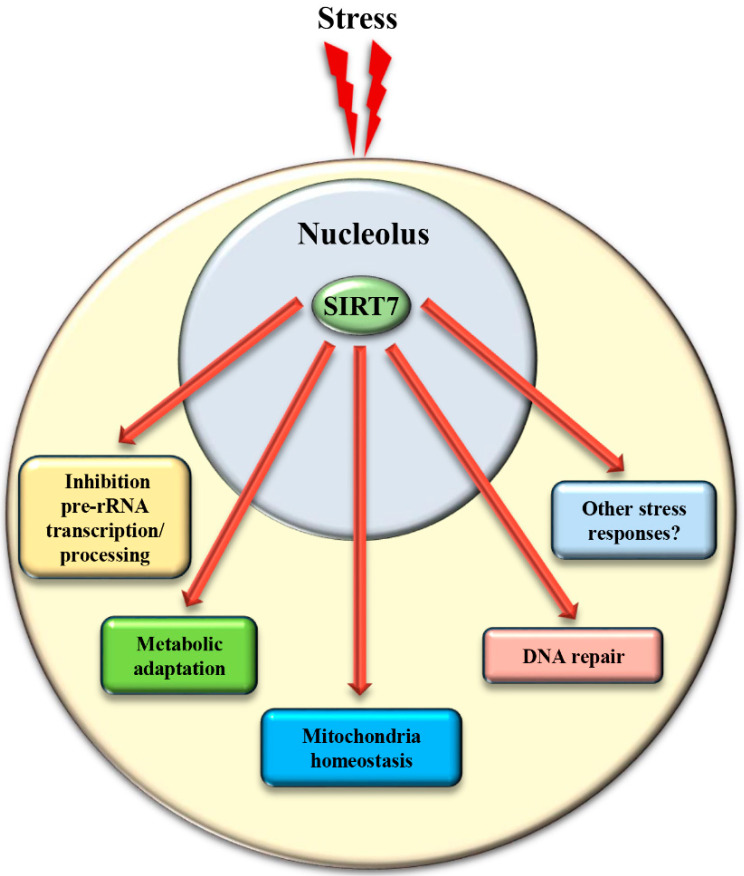
Translocation of *SIRT7* from the nucleolus following stress is a critical event to activate *SIRT7*-regulated extra-nucleolar functions. See text for details.

**Table 1 genes-12-01361-t001:** Table summarizing *SIRT7* nucleolar functions under physiological and stress conditions.

*SIRT7* Nucleolar Functions		
Function	Cell Type	Condition
Stimulation of rDNA transcription and ribosome biogenesis	Cancer cell lines and human embryonic kidney cells	Physiological conditions [27,28,29,30,31].
Maintenance of rDNA repeats integrity	Primary mouse embryonic fibroblasts (MEFs), mouse liver, human fibroblast-like fetal lung cells	Physiological conditions [22,47].
Resolution of R-loops	Cancer cell lines and human embryonic kidney cells	Physiological conditions [53].
Stabilization of p53 through NPM deacetylation	Cancer cell lines, MEFs, mouse skin	UV irradiation [17]
Stabilization of p53 through MDM2 degradation	Cancer cell lines	glucose starvation [59]
Inhibition of pre-rRNA transcription following stress	Cancer cell lines and human embryonic kidney cells	Nutrient stress [28,66]. hypertonic stress [31].

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
