# Peer review of "SIRT7 Acts as a Guardian of Cellular Integrity by Controlling Nucleolar and Extra-Nucleolar Functions"

_genes, 2021, doi:10.3390/genes12091361_

Round 1
Reviewer 1 Report
Sirtuins are a group of NAD+-dependent deacetylases, which counteract with HATs that conduct lysine/histone acetylation. Sirtuins also counteract with ADP-ribosylation enzymes, which use NAD+ as a substrate. Under physiological and stress conditions, Sirtuins directly modify protein activity, functionality and location, thereby driving the cell fate, via for example transcriptional regulation of target genes.
In mammals, there are seven members in the Sirtuin family. While Sirt1 is perhaps the most studied member, the functions of other Sirtuins are still under debate. The specialty of Sirt7 is its enrichment in nucleoli (in addition to its nuclear location) and thus involved in protein production, stress response, genomic stability and nutrition-related metabolism, malfunctioning of which can cause human diseases, including aging.
The authors of the manuscript have summarized so-far known functions of Sirt7. The authors provide a comprehensive, albeit a brief, review on the function of Sirt7. The information is very helpful for the field of, not only Sirtuins and rDNA/rRNA metabolism, but also nutrition biology. The manuscript reads well.
The following specific comments should be addressed:
- The title focuses on the role of Sirt7 in the “nucleolus”. However, there is a significant proportion of discussion on other functions beyond the nucleolus (entire Session #3), which is indeed useful to be included. Thus the title should be modified accordingly.
- Fig 1. It would be even more helpful to general readers to show a domain structure of Sirt7, which was mentioned only in the main text.
- Fig 1 (c). The flow or direction (upstream and downstream) of Sirtuins’ function needs a better description in the legend. What does the color code (red and green) stand for?
Author Response
Reviewer 1.
Sirtuins are a group of NAD+-dependent deacetylases, which counteract with HATs that conduct lysine/histone acetylation. Sirtuins also counteract with ADP-ribosylation enzymes, which use NAD+ as a substrate. Under physiological and stress conditions, Sirtuins directly modify protein activity, functionality and location, thereby driving the cell fate, via for example transcriptional regulation of target genes.
In mammals, there are seven members in the Sirtuin family. While Sirt1 is perhaps the most studied member, the functions of other Sirtuins are still under debate. The specialty of Sirt7 is its enrichment in nucleoli (in addition to its nuclear location) and thus involved in protein production, stress response, genomic stability and nutrition-related metabolism, malfunctioning of which can cause human diseases, including aging.
The authors of the manuscript have summarized so-far known functions of Sirt7. The authors provide a comprehensive, albeit a brief, review on the function of Sirt7. The information is very helpful for the field of, not only Sirtuins and rDNA/rRNA metabolism, but also nutrition biology. The manuscript reads well.
The following specific comments should be addressed:
R1. We thank the reviewer for the positive evaluation of our manuscript. We provide a point-by-point response to the reviewer´s comments:
- The title focuses on the role of Sirt7 in the “nucleolus”. However, there is a significant proportion of discussion on other functions beyond the nucleolus (entire Session #3), which is indeed useful to be included. Thus the title should be modified accordingly.
R2. We thank the reviewer for the suggestion. We have changed the title of the manuscript, which now reads: “SIRT7 acts as a guardian of cellular integrity by controlling nucleolar and extra-nucleolar functions.”
- Fig 1. It would be even more helpful to general readers to show a domain structure of Sirt7, which was mentioned only in the main text.
R3. We thank the reviewer for the constructive comment. We indicated now the nuclear and nucleolar localizing sequences (NLS and NoLS, respectively) in the revised Fig. 1b and changed the relative figure legend.
- Fig 1 (c). The flow or direction (upstream and downstream) of Sirtuins’ function needs a better description in the legend. What does the color code (red and green) stand for?
R4. We thank the reviewer for the constructive comment. We indicated the color code of Fig. 1c in the figure legend.
Reviewer 2 Report
Please find my comments in a file attached.

Author Response
Reviewer 2
Sirtuins are at the hot spot of modern biomedicine. The functions of sirtuins, especially sirtuin 7, are well established, including the role in the pathogenesis of certain diseases. However, the mechanisms of action of the sirtuins at the subnuclear level have not been summarized sufficiently. Therefore, the review proposed by the authors is undoubtedly of considerable interest.
The MS makes a very favorable impression. The material is presented in a logical, understandable and well-illustrated manner. I have no significant comments and questions after reading. In this regard, the following comments are purely advisory in nature.
R1. We thank the reviewer for the constructive and positive evaluation of our manuscript.
- From my point of view, it would be helpful to add a table that summarizes the nucleolar effects of Sirt7. This table should include information on the cell types that exhibit the Sirt7 effect. The proposed table can illustrate the degree of universality of the observed effects without additional reading of the original articles.
We thank the reviewer for the constructive comment, which prompted us to add a table (table 1) in the revised manuscript illustrating the nucleolar effects of SIRT7 under physiological and stress conditions.
- At the beginning of Section 2.1, I suppose it would be useful to add some additional links to current reviews that describe structure and function of the nucleoli under physiological conditions. In the MS, the authors have cited only one review ([25]), which focuses on the process of ribosome biogenesis in cancer cells.
We thank the reviewer for the suggestion. We introduced an additional citation to a review describing structure and function of the nucleolus under physiological conditions.
- Technical points
3.1. Line 201: the specific name is written with a lowercase letter (should be Drosophila melanogaster).
The reviewer is right. We apologize for the shortcoming. We have corrected this mistake.
3.2. In Fig. 4 (blue block) typo: apparently, there should be “homeostasis” instead of “homeostais”.
We apologize for the shortcoming. We have corrected this mistake
3.3. Line 389: should be “telangiectasia” (now “talangectasia”).
We apologize for the shortcoming. We have corrected this mistake